# A Media Analysis of the COVID-19 Tobacco Sales Ban in South Africa

**DOI:** 10.3390/ijerph20186733

**Published:** 2023-09-07

**Authors:** Chengetai Dare, Nicole Vellios, Praveen Kumar, Radhika Nayak, Corné van Walbeek

**Affiliations:** 1Research Unit on the Economics of Excisable Products, School of Economics, University of Cape Town, Private Bag X3, Rondebosch 7700, South Africa; nicole.vellios@uct.ac.za (N.V.); cornelis.vanwalbeek@uct.ac.za (C.v.W.); 2Department of Commerce, Manipal Academy of Higher Education, Manipal 576104, Karnataka, India; praveen.kumar@manipal.edu; 3Department of Community Medicine, Kasthurba Medical College, Manipal Academy of Higher Education, Manipal 576104, Karnataka, India; nayak.radhika@manipal.edu

**Keywords:** media analysis, COVID-19, tobacco sales ban, lockdown, illicit trade, tax revenue

## Abstract

The South African government introduced a nationwide lockdown in March 2020 to mitigate the spread of COVID-19. Among other restrictions, the government banned the sale of tobacco products. The ban lasted for nearly five months. We performed a Google search using the keywords smok*, puff*, lockdown, tobacco, and cigarette* for articles published in English from 23 March 2020 to 18 December 2020. This yielded 441 usable online media articles. We identified and categorised the main arguments made by proponents and opponents of the tobacco sales ban. Three themes were identified: medical, legal, and economic/financial. Legal aspects were covered in 48% of articles, followed by economic (34%), and medical aspects (18%). The media was generally ambivalent about the tobacco sales ban during the first five weeks of lockdown. Sentiment subsequently turned against the ban because the medical rationale was not well communicated by the government. There was limited empirical evidence of a link between smoking and contracting COVID-19, and the sales ban was ineffective since most smokers still purchased cigarettes. Policy framing in the media plays an important role in how the public receives the policy. Any future tobacco control policy intervention should be better considered, especially within the context that cigarettes are easily accessed on the illicit market in South Africa.

## 1. Introduction

South Africa recorded its first COVID-19 case on 5 March 2020. President Ramaphosa declared a National State of Disaster on 15 March 2020. On 23 March 2020, the President announced a nationwide lockdown, effective 27 March 2020. On 25 March, Dr. Dlamini-Zuma (Minister of Co-operative Governance and Traditional Affairs) published regulations on the national lockdown [1]. As part of the lockdown measures to mitigate the spread of COVID-19, non-essential economic sectors (including tobacco) were prohibited from producing and trading their goods.

On 23 April 2020, the President announced that the tobacco sales ban would end on 1 May 2020. A few days later, Minister Dlamini-Zuma, who, together with the President, chaired the National Coronavirus Command Council (NCCC), announced that the ban would not end on 1 May 2020.

The continuation of the tobacco sales ban was controversial. Finance Minister Mboweni announced that he was against continuing the ban (Articles 56; 75; 121). Various smoking lobby groups held public protests. The tobacco industry filed two lawsuits against Minister Dlamini-Zuma. The industry lost the first lawsuit, which was primarily based on the argument that tobacco products are addictive and should not be banned. The industry succeeded with the second lawsuit, in which they argued that the ban was unconstitutional because, among other things, it undermined people’s right to dignity and prevented people in the tobacco value chain from earning a living. Minister Dlamini-Zuma appealed the court’s finding that the ban was unnecessary, but the Supreme Court of Appeal rejected the appeal in June 2022. The Supreme Court of Appeal ruled that there was no scientific justification for the continued sales ban beyond the initial period, as there was no evidence that short-term quitting has clinical significance for COVID-19 severity and outcomes. 

The tobacco sales ban in South Africa drew much attention. Several surveys investigating smokers’ behaviour were conducted during the sales ban [2,3,4,5]. Based on these surveys and other data, several papers were published, highlighting the varied implications of the sales ban [4,6,7,8,9]. Despite the ban, the sale of cigarettes did not cease; in fact, the ban entrenched South Africa’s already large illicit market [4] Filby, van der Zee, and van Walbeek [6] conclude that for a sales ban to be effective, (1) the illicit market must be under control before implementing a sales ban, and (2) an effective sales ban needs to be synchronised with a ban on the manufacture, transport, and distribution of cigarettes. These two supply-side factors were not in place in South Africa, which resulted in an increase in illicit trade. For any policy intervention, how the intervention is framed in the media plays an important role in how the public receives the policy. The media can enhance or undermine a public health intervention [9].

Over the recent past, media analysis has been widely carried out in tobacco control research [10,11,12,13,14]. In this study, we analyse the main arguments made by proponents and opponents of the tobacco sales ban. We follow the approach of Hilton, Wood, Patterson, and Katikireddi [15], who analysed the arguments made by ‘key claim-makers’ regarding the implementation of a minimum alcohol unit price in Scotland based on a media analysis of 262 articles from ten UK national newspapers. The study examined advocates’ and critics’ constructions of the alcohol problem and whether a minimum unit price would address the issue. They identified categories for a coding frame around a priori research questions.

Findings from our study will help policymakers in future tobacco control policy interventions. To the best of the authors’ knowledge, this will be the second study after Egbe, Ngobese, Barca, and Crosbie [9] to examine media coverage of the tobacco sales ban in South Africa. This study places greater emphasis on the economic aspects of the sales ban and covers a somewhat longer period, from 23 March 2020 to 18 December 2020. 

## 2. Data and Methods

Focusing on South African media houses, we performed a Google search using the keywords smok*, puff*, lockdown, tobacco, and cigarette* for articles published from 23 March 2020 (when the sales ban was announced) to 18 December 2020 (a week after the High Court ruled that the ban was unnecessary and unconstitutional). We started by scraping media reports from Google News using Python, then scraped the largest news websites, and finally smaller news outlets. The search yielded 1512 potentially useful articles. We excluded duplicate articles (n = 263), video articles (n = 386), and articles from non-national media outlets (n = 422). The final sample consisted of 441 articles (see Appendix A).

First, the articles were randomly divided into three groups and read by three researchers (CD, PK, and RN). The articles underwent a rigorous analysis process and were systematically analysed using an inductive qualitative content analysis approach. The researchers read each article multiple times to identify relevant textual segments related to the research questions. Next, texts were assigned to codes with descriptive labels, and as more related texts emerged, new codes were created or added to existing ones. The first round of coding was conducted manually to develop the initial codebook. PK and RN initially conducted the data coding with input from CD. The codes were then carefully examined for coherence and clarity and to identify any relationships among them. The research team engaged in discussions to refine the codes and create subcodes through repeated readings of the articles. Subsequently, the articles were systematically coded in the second reading using NVivo, based on the developed code list. A codelist contained code names and definitions for each code and was used to allow consistency in coding across articles and thus ensure quality data analysis.

Finally, the codes were further organised into major themes and sub-themes, grouping them based on similarities and differences. To ensure accuracy, the themes and sub-themes were compared with the textual segments of the articles, validating that they accurately reflected the message conveyed. The interpretations (codes, themes, and sub-themes) were exchanged among the team for validation purposes. In cases of discrepancies, authors would discuss them to reach a consensus. The coded data were then used to develop a narrative analysis. 

## 3. Results

Of the 441 articles, most were from Eye-Witness News (EWN) (n = 113), the Independent Online (IOL) (n = 81), the Daily Maverick (n = 65), and the South African Broadcasting Corporation (SABC) News (n = 49) (Figure 1). 

Throughout the sales ban period (Figure 2, grey band: 25 March–17 August 2020), there was substantial media attention on tobacco issues compared to the post-ban period. Some weeks attracted more media attention than others. These spikes in media attention were driven mostly by legal events, e.g., the tobacco industry initiating legal action and the cases being heard in court. 

The main players (or organisations) most frequently mentioned are shown in Table 1. The percentages in the last row represent the frequency with which the stakeholder is mentioned. President Ramaphosa (60%) and Minister Dlamini-Zuma (50%) were mentioned most frequently, followed by the NCCC (16%), political parties (5–11%), and the Finance Minister (10%). The South African Revenue Service (SARS) was mentioned in 21% of the articles, and the High Court in 27% of the articles. The Fair-trade Independent Tobacco Association (FITA), a body that represents the smaller tobacco companies, was mentioned in 40% of the articles, whereas British American Tobacco South Africa (BATSA) was mentioned in 27% of the articles. 

Eight percent of the articles mentioned research on the impact of the sales ban conducted by the Research Unit on the Economics of Excisable Products (REEP), while 5% of the articles mentioned research on the impact of the sales ban conducted by the Human Sciences Research Council (HSRC).

We identified three themes, each with several distinct sets of arguments: medical, legal, and economic/financial (Table 2). The primary theme is the most important, and the secondary theme (where applicable) is less important. If an article reported mostly on the legal aspects of the ban but covered medical aspects to a lesser extent, we classified the article as having the legal aspects as the primary theme and the medical aspects as a secondary theme.

The legal aspects of the sales ban were the primary theme of 48% of articles, followed by economic aspects (34%), and medical aspects (18%). The economic aspects of the sales ban received much attention as a secondary theme. Considering the primary and secondary themes together, the economic aspects are mentioned in 56% of articles, followed by the legal aspects (52%), and the medical aspects (22%). Among the top ten publishing houses listed in Table 2, the legal and economic aspects received the most attention, while, with one exception (Fin24), the medical aspects received the least attention.

### 3.1. Medical Theme

The most common argument against the sales ban, typically made by the tobacco industry and pro-smoking groups, was that the ban was unsupported by scientific evidence (Articles 11; 58; 367; 384). In its court case against the government, FITA challenged the scientific evidence linking smoking and COVID-19 severity, saying that ‘*more peer-reviewed studies would have to be done to support the view that smokers are more likely to contract severe cases of COVID-19, as the current evidence is inconclusive*’ (Article 306). FITA also critiqued the World Health Organisation’s (WHO) assertion that linked smoking to COVID-19 infections, saying ‘*the WHO’s statement was inconclusive as it did not have enough information to study any possible link between tobacco and prevention of COVID-19’* (Article 368).

Several articles reported on preliminary research from China and Italy that suggested that smokers were disproportionately under-represented among COVID-19 patients (Articles 221; 292; 311; 373) and that smoking might have a protective effect against COVID-19 (Articles 221; 311). The view that the medical evidence did not support the sales ban was shared by some prominent medical professionals. The deputy director of the National Institute for Communicable Diseases indicated that there is ‘*no direct or very good information*’ that cigarettes increase one’s chances of contracting COVID-19 (Article 365). It was argued that there is no evidence that smoking increases COVID-19 transmission (Article 441).

The continuation of the sales ban after the initial five weeks was rationalised by Minister Dlamini-Zuma on the grounds that smokers tend to smoke in groups, sharing cigarettes and thus spreading COVID-19 through saliva (Articles 211; 266). This rationale was ridiculed by the media (Articles 68; 171; 344; 349). Several articles noted that it is unrealistic to expect that accumulated lung damage over many years could be suddenly undone (Articles 260; 268; 313; 439). One of South Africa’s top scientists (Professor Shabir Madhi) said that ‘*the damage caused by smoking occurs over a long period, and a short period of not smoking cannot undo long-term damage already done*’ (Article 253), making the ban a bad idea.

Some articles reported that the government was ignoring the mental health aspects of a sudden nicotine withdrawal, which include ‘*severe cravings, irritability and anxiety*’ (Articles 313; 331). Because of nicotine’s addictiveness, it was argued that smokers would be forced to smoke more harmful products as alternatives (Article 363). A spokesperson for a large hospital said the ban caused ‘*those manufacturing their own cigarettes to take chances with ingredients that could leave consumers sick—or even dead*’ (Article 363). She indicated that, as a hospital, ‘*they have had over 50 patients brought in for cigarette experiment cases, including using ingredients such as tea leaves*’ (Article 363).

In contrast, Minister Dlamini-Zuma insisted that the ‘*ban was necessary to protect public health and decrease the potential strain on the country’s health system from COVID-19’* (Article 209). She argued that smokers were at a higher risk of contracting COVID-19 due to hand movement to the mouth when smoking cigarettes. The media did not report on any government initiatives to help smokers quit tobacco.

The ban was supported by the HSRC, which argued in the early weeks of the lockdown that ‘if only 1 percent of the 8 million smokers were to contract COVID-19, this means 80,000 smokers would be infected. If an estimated 5 percent were to need ICU, this would translate to about 4000 people needing ICU hospital beds and ventilators. Under current calculations, this would exceed the availability of ventilators and place health workers at risk’ (Article 327).

The government’s position was backed by the South African Thoracic Society, the Cancer Association of South Africa, the Heart and Stroke Foundation of South Africa, and the National Council Against Smoking (Articles 60; 264).

### 3.2. Legal Theme

The government banned the sale of tobacco products because they were not classified as essential items in the Disaster Management Act 2020, which implemented the lockdown [1]. Tobacco products were classified as non-essential because they do not ‘*by their nature, fall into the same category as goods which are life-sustaining or necessary for basic functionality*’ (Article 426). However, there were some contradictions in the interpretation of the ban, especially at the outset. The Western Cape (one of South Africa’s nine provinces) allowed the sale of tobacco products and announced that they ‘*may be sold during the lockdown, but only together with essential goods*’ (Article 420). The Police Minister corrected the misinterpretation, emphasising that the Western Cape government could not implement provincial regulations that differed from national regulations (Article 178; 240).

Two legal challenges were made against the tobacco sales ban, both initiated by the tobacco industry. The first legal appeal was made by FITA in early May 2020, when the government failed to lift the sales ban after initially indicating that it would. FITA argued that cigarettes should be regarded as essential products because they are addictive (Article 261; 384). FITA’s application was supported by members of a Facebook group, Smokers Unite One, which, at the time, had 289,000 members (Article 266). About six weeks later, FITA’s case was rejected by the North Gauteng High Court on the basis that cigarettes and related tobacco products do not fall into the same category as goods that are ‘*life sustaining or necessary for basic functionality*’ (Article 189). The court held that, even if a substance is addictive, that does not necessarily mean it is essential. FITA appealed the judgement, accusing the court of misinterpreting the Disaster Management Act (Article 424). FITA lost the appeal, and the ban was upheld (Article 425). In its ruling, the High Court said that ‘*the association [FITA] failed to show why an appeal should be heard*’ (Article 111).

In June 2020, the government relaxed some of the COVID-19 restrictions but kept the tobacco sales ban in place. Eye Witness News cited BATSA, which indicated that it ‘*has made every effort to constructively engage with the government since the ban came into force, including making detailed submissions, along with other interested parties, to various ministers, as well as directly to the Presidency. To date, no formal response has been received from the government. BATSA has also not been included in any of the government’s consultation processes so far*’ (Article 208). This led BATSA to file a lawsuit against the government at the Western Cape High Court (Articles 208; 308). The lawsuit was supported by other multinational companies (Japan Tobacco International and Philip Morris South Africa), as well as those in the tobacco value chain (such as the Black Tobacco Farmers Association and the South African Tobacco Transformation Alliance). The court hearings were delayed or postponed, resulting in accusations and counteraccusations between BATSA and the government. Independent Online reported that the ‘*giant tobacco producer [BATSA] and the government are pointing fingers at each other over the delay of another case that will determine whether the ban on the sale of tobacco products should continue*’ (Article 260). BATSA condemned the delay, saying that ‘*this delaying of justice and a resolution of this issue is inexplicable*’. On the other hand, the government accused BATSA of having caused the postponement by submitting new evidence without timely informing them (Article 260).

When the court proceedings started in August 2020, BATSA argued that the ban was unconstitutional and violated a series of fundamental rights, including the right to dignity (Section 10 of the Constitution), which extends to autonomy and the ability to make choices (Articles 308; 373). BATSA also argued that the ban violated the right to privacy, the right to bodily and psychological integrity, and the rights of tobacconists and tobacco farmers to practice their trade (Articles 308; 373). In response, the government insisted that the ban was a temporary measure intended to save lives (Article 134; 152) and cited the HSRC’s claim that ‘*if all the country’s smokers quit, this would potentially free up 4000 hospital beds*’ (Article 270). BATSA challenged the HSRC’s numbers on the impact that smoking COVID-19 patients would have on the number of beds occupied, based primarily on the fact that surveys had shown that, at most, about 1 million smokers had quit smoking during the sales ban period (Articles 270; 368). BATSA argued that, ‘*if the minister realistically expects only about 1 million smokers to quit, this would free up 500 hospital beds over the duration of the COVID-19 pandemic*’, which is much lower than the HSRC’s estimates (Articles 355; 368). The court handed down its ruling on 11 December 2020 in BATSA’s favour, saying that the ban was unnecessary and unconstitutional (Article 251; 339). However, at this point, the judgement was moot as the tobacco sales ban had already been lifted on 18 August 2020.

The legality of the sales ban was also challenged by various other people and groups, including the SARS Commissioner and hundreds of thousands of people who signed a petition calling for an end to the sales ban (Articles 56; 121). However, members of the public did not file legal challenges.

### 3.3. Economic/Financial Theme

The groups that made economic/financial arguments about the sales ban were all strongly opposed to it. The arguments were raised by tobacco manufacturers, tobacco farmers, retailers, the Finance Minister, the SARS Commissioner, smokers, and some academics. The arguments were centred on three aspects: foregone tax revenue, the illicit market, and the livelihoods of people in the tobacco value chain.

#### 3.3.1. Foregone Tax Revenue

The industry argued that the ban resulted in forgone government revenue. FITA indicated that the government was losing about R1.5 billion (US$ 90 million) monthly on excise taxes; this figure would be greater if other taxes, like value-added tax, were included (Articles 239; 240). BATSA and Tax Justice South Africa (TJSA), a group closely aligned with BATSA, said the government was losing R35 million daily in foregone excise taxes (Article 259).

The SARS Commissioner highlighted that the sales ban greatly reduced SARS’s excise tax revenue (Articles 32; 56; 240).

#### 3.3.2. The Illicit Tobacco Market

Because of the sales ban, all cigarette sales were illicit. Smokers accused the government of turning them into criminals (Article 195). Hundreds of thousands of opponents to the sales ban signed a petition against it (Article 47), and smokers protested across the country, including at Parliament (Article 195).

During the first five weeks of the lockdown, most factories were shut down, including cigarette factories. Towards the end of the initial five-week period, there were reports that at least one cigarette factory (which turned out to be BATSA) had been producing cigarettes despite the ban (Article 54). Subsequently, the government allowed cigarette manufacturers to produce cigarettes for export (Article 76). However, it was reported that at least two-thirds of cigarettes earmarked for export were not exported (Article 320). These cigarettes were illegally sold in South Africa.

Several articles reported on the arrests of illicit traders and/or the confiscation of cigarette consignments, especially from Zimbabwe (Articles 60; 76; 187). One of the biggest busts involved a R7 million ($425,000) consignment of illegal cigarettes from Zimbabwe (Article 58).

The media reported on several surveys of smokers’ behaviour conducted during the sales ban. The first was a HSRC survey, conducted from 9–16 April 2020 (i.e., 2–3 weeks into the sales ban). The HSRC found that 12% of smokers purchased cigarettes after sales were banned (Article 355).

The Research Unit on the Economics of Excisable Products (REEP) conducted two surveys during the sales ban [2,3]. Findings from the first survey (conducted from 29 April to 11 May 2020) indicated that 16% of pre-ban smokers had quit during the sales ban [2]. However, 91% of continuing smokers had purchased cigarettes illicitly during the lockdown (Articles 17; 240). On average, the prices paid by smokers were 90% higher than before the sales ban (Articles 17; 41; 235). In July 2020, the media reported on a second REEP survey (conducted from 4–19 June 2020) (Articles 172; 272; 277). The results were qualitatively similar, but the average price of cigarettes was nearly 250% higher than pre-lockdown prices (Articles 68; 172; 272).

#### 3.3.3. Livelihoods

Retail outlets wrote an open letter to the government highlighting the ‘devastation’ caused by the tobacco ban and urging them to end it (Article 182). The letter was backed by the TJSA, which regarded the letter ‘*as a timely reminder of how the tobacco ban was destroying the livelihoods of honest workers while enriching criminals in the illicit tobacco trade*’ (Article 267).

The Black Tobacco Farmers Association and the South African Informal Traders Alliance (SAITA) also appealed to the government to lift the ban as their members were ‘*on the brink of extinction*’ (Article 121). Limpopo Tobacco Processors, the largest supplier of tobacco leaf to domestic buyers in South Africa, and the South Africa Tobacco Transformation Alliance (SATTA), an alliance of black tobacco farmers, tobacco processors, and BATSA, also called for the ban to be lifted (Article 44). According to SATTA, ‘*if the ban is not lifted soon, more than 150 black tobacco farmers who were struggling to make a living in rural South Africa would go out of business*’ (Article 121). Smaller businesses selling e-cigarettes reported an ‘*existential crisis which is likely to lead to widespread bankruptcies and unemployment*’ (Article 335).

The view that the sales ban had a negative impact on employment was also voiced by the SARS Commissioner (Article 56) and by the Democratic Alliance (DA), the official opposition party in South Africa (Article 205). The DA said it would challenge the ban in court (Article 205), but this did not materialise.

## 4. Discussion

The aim of the study was to analyse the main arguments made by proponents and opponents of the tobacco sales ban, as reported in the media. We found that the tobacco sales ban was covered extensively. The legal and economic aspects of the sales ban received the most media attention. On these two aspects, the media generally reported negatively on the sales ban. For the medical theme, the reporting was more balanced, with reports often quoting organisations that supported the sales ban.

At the beginning of the sales ban, the media was mostly ambivalent about it, given the uncertainty that surrounded COVID-19. After the sales ban was extended beyond five weeks, the public and the media became more vocal in their opposition. Support for the sales ban may have decreased because: (1) the medical rationale received little empirical support; and (2) the sales ban was flouted. Public and media support for the tobacco sales ban was also undermined because of a perception that the government’s overall handling of the epidemic was harsh, uncaring, and inconsistent (Articles 194; 204; 332). Some interventions and restrictions (not only on tobacco) received a lot of negative exposure (Article 304; 437). As the ‘face’ of the government’s COVID-19 response, Minister Dlamini-Zuma received much criticism.

Towards the end of the sales ban, the government’s rationale for the sales ban focused less on the relationship between smoking and COVID-19 than on the negative consequences of smoking per se (Articles 94; 193; 436). The High Court and the Supreme Court of Appeal (Article 251; 339) argued that while smoking is undoubtedly harmful, this is not a sufficient reason to ban its sales, especially in the context of a stressful period for the whole population.

The ban failed to achieve its goal of preventing people from smoking but rather entrenched the illicit tobacco market. Reports by the media that the sales ban increased illicit trade are in line with findings by van Walbeek, Filby, and van der Zee [4], who found that a substantial proportion of smokers continued purchasing illicit cigarettes after the ban ended. A recent study by Vellios [16] estimated that the illicit market in 2021, i.e., well after the sales ban has been lifted, comprised nearly 54% of the market, up from between 30% and 35% in 2017 [17]. FITA-affiliated tobacco companies greatly expanded their market share and made substantial profits during the sales ban period [4], but were publicly strongly opposed to the ban (evident by the relatively large number of times that they were cited in the media and because they launched the first court case). On the other hand, the multinational companies lost market share during the sales ban and even afterwards [4].

Although the ban was well-intended when it was introduced, its continual extension caused the government to lose a significant amount of revenue. The estimates of the revenue lost varied between the different tobacco industry groups. TISA claimed that R1.5 billion in excise revenue was lost monthly, compared to BATSA’s and TJSA’s claims of R1.05 billion lost monthly (Articles 239; 240; 259).

The sales ban failed to meet its objectives, resulting in the government’s reputation being undermined. As Egbe, Ngobese, Barca, and Crosbie [9] pointed out, it is likely to be more difficult to get the public to support future tobacco-control legislation.

### Strengths and Limitations

This study employed code verification to ensure the themes used were not biased towards one researcher. The period covered in this paper was crucial for understanding issues related to the ban. However, the study has limitations. We did not include international media, YouTube videos, or social media, which may reduce the scope of our data. Our study was limited to media articles published in English, so we may have missed articles written in other local languages. It would be important to examine if the sample size would increase if data were to be collected using a different search engine, e.g., Google Trends.

## 5. Conclusions

The tobacco sales ban aspect of South Africa’s lockdown was regularly covered by the media. Opinions varied significantly, and stakeholders expressed polarised views, framed mostly on three themes: medical, legal, and economic. The framing of a policy intervention in the media plays an important role in how the public receives the policy. When the sales ban was first introduced, there was uncertainty about whether or not it was a good idea. As the weeks went on, the sentiment generally turned strongly against the sales ban. At a practical level, the sales ban was not working because most smokers were able to purchase cigarettes on the illicit market. Although smokers need support when quitting, the South African government did not provide this. Had the government provided any smoking cessation support services, the sales ban might have received more support from the media and the public. Any future tobacco control policy intervention should take cognizance of the prevalence of illicit cigarettes and address the illicit market. Furthermore, appropriate cessation support services should be offered to smokers who wish to quit tobacco.

## Figures and Tables

**Figure 1 ijerph-20-06733-f001:**
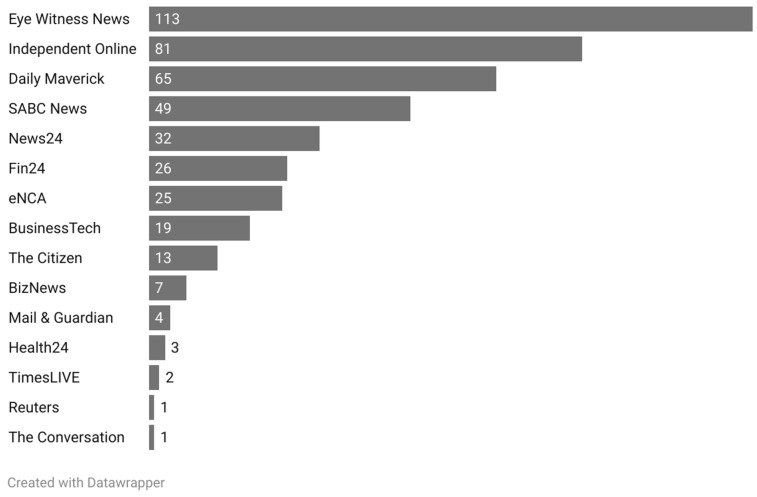
Distribution of articles by news media outlet.

**Figure 2 ijerph-20-06733-f002:**
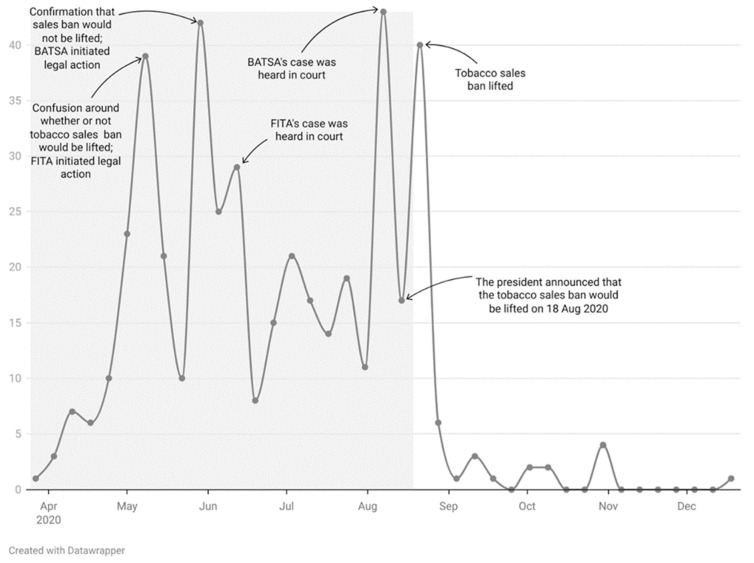
Number of articles published per week from 27 March to 12 December 2020.

**Table 1 ijerph-20-06733-t001:** Number of times stakeholders are mentioned in the news articles (n = 441).

		Politicians, Political Parties and Political Bodies							Government Bodies		Tobacco Industry		Research Institutions	
Publishing House	Total	President Ramaphosa	Minister of Co-operative Governance and Traditional Affairs (Dlamini-Zuma)	National Coronavirus Command Council (NCCC)	African National Congress (ANC)	Democratic Alliance (DA)	Economic Freedom Fighters (EFF)	Minister of Finance (Tito Mboweni)	South African Revenue Service (SARS)	High Court	Fair Trade Independent Tobacco Association (FITA)	British American Tobacco South Africa (BATSA)	Research Unit on the Economics of Excisable Products (REEP)	Human Sciences Research Council (HSRC)
Eye Witness News	113	63	49	17	9	9	6	12	21	31	40	29	7	6
Independent Online	81	56	49	9	4	2	1	5	14	27	38	31	7	3
Daily Maverick	65	47	39	13	19	12	7	13	19	15	26	19	4	5
SABC News	49	31	29	15	12	7	9	9	19	19	24	17	9	5
News24	32	23	17	0	0	0	0	0	0	2	1	2	0	0
Fin24	26	14	16	6	2	1	0	2	5	2	16	8	0	2
eNCA	25	8	5	3	0	0	0	0	1	8	16	3	0	2
BusinessTech	19	8	5	3	1	0	0	1	6	5	5	7	3	0
The Citizen	13	6	5	2	1	2	0	1	2	4	3	3	2	0
BizNews	7	3	2	0	0	0	0	1	2	3	2	1	0	0
Mail & Guardian	4	1	2	0	0	0	0	0	2	2	1	2	2	0
Health24	3	3	2	1	1	1	1	1	0	1	0	1	0	0
TimesLIVE	2	0	2	0	0	0	0	0	0	2	2	0	0	0
Reuters	1	1	0	0	0	0	0	0	0	0	1	1	0	0
The Conversation	1	1	0	0	0	0	0	0	0	0	0	0	0	0
Total	441	265	222	69	49	34	24	45	91	121	175	124	34	23
Percentage		60%	50%	16%	11%	8%	5%	10%	21%	27%	40%	28%	8%	5%

**Table 2 ijerph-20-06733-t002:** Number of articles identified in each theme.

		Primary Theme			Percentage Primary Theme			Percentage (Primary and Secondary Theme)		
Publishing House	Total	Medical	Legal	Economic	Medical	Legal	Economic	Medical	Legal	Economic
Eye Witness News	113	18	50	45	16%	44%	40%	23%	50%	63%
Independent Online	81	9	48	24	11%	59%	30%	14%	62%	53%
Daily Maverick	65	17	21	27	26%	32%	42%	28%	40%	58%
SABC News	49	8	27	14	16%	55%	29%	18%	55%	55%
News24	32	6	17	9	19%	53%	28%	22%	63%	53%
Fin24	26	9	10	7	35%	38%	27%	54%	42%	46%
eNCA	25	1	21	3	4%	84%	12%	4%	84%	48%
BusinessTech	19	5	6	8	26%	32%	42%	26%	37%	63%
The Citizen	13	1	8	4	8%	62%	31%	15%	77%	38%
BizNews	7	0	1	6	0%	14%	86%	14%	29%	86%
Mail & Guardian	4	1	0	3	25%	0%	75%	25%	25%	100%
Health24	3	2	1	0	67%	33%	0%	67%	33%	0%
TimesLIVE	2	0	2	0	0%	100%	0%	0%	100%	0%
Reuters	1	0	1	0	0%	100%	0%	0%	100%	0%
The Conversation	1	1	0	0	100%	0%	0%	100%	0%	0%
Total	441	78	213	150	18%	48%	34%	22%	54%	56%

## Data Availability

This study used publicly available news articles. The list of news articles used is attached as Appendix A.

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
