# Peer review of "A Media Analysis of the COVID-19 Tobacco Sales Ban in South Africa"

_ijerph, 2023, doi:10.3390/ijerph20186733_

Round 1
Reviewer 1 Report
The current study conducted an analysis of online articles about a tobacco sales ban in South Africa during the COVID-19 lockdown from March-December 2020. While I believe the paper reports findings from an important topic and will be relevant to a global population, I have concerns about the methods used to collect, analyze, and report the findings. Overall, the paper is well-written, and if methodological concerns are addressed and better described, I believe this paper will provide an important contribution to literature.
Introduction
· Line 53-54 – Instead of saying “several papers were published”, what did these studies? What was relevant about their findings for the current study?
· While the first paragraphs do a nice job of providing context for the timeline of events, the final two paragraphs fall short in explaining why online media posts a relevant source of data are to examine this phenomenon. Provide at least one paragraph explaining why is it important to examine media data about a tobacco control policy after implementation.
· The authors appear to be citing mainly their own articles. I encourage the authors to expand their literature search to better describe the relevant and importance of using media data. Research has explored trends of tobacco coverage in more traditional media channels, such as newspapers and television:
o Nelson, D. E., Pederson, L. L., Mowery, P., Bailey, S., Sevilimedu, V., London, J., ... & Pechacek, T. (2015). Trends in US newspaper and television coverage of tobacco. Tobacco Control, 24(1), 94-99.
· Prior research has also used google trends data to explore a myriad of topics within tobacco control, including interactivity on tobacco control websites, cessation seeking, and product popularity.
o Freeman, B., & Chapman, S. (2012). Measuring interactivity on tobacco control websites. Journal of health communication, 17(7), 857-865.
o Troelstra, S. A., Bosdriesz, J. R., De Boer, M. R., & Kunst, A. E. (2016). Effect of tobacco control policies on information seeking for Smoking Cessation in the Netherlands: a Google Trends Study. PLoS One, 11(2), e0148489.
o Tabuchi, T., Fukui, K., & Gallus, S. (2019). Tobacco price increases and population interest in smoking cessation in Japan between 2004 and 2016: a Google trends analysis. Nicotine and Tobacco Research, 21(4), 475-480.
o Troelstra, S., Bosdriesz, J., de Boer, M., & Kunst, A. (2014). Effect of tobacco control policies on information seeking for smoking cessation in the Netherlands: A Google Trends study: Jizzo Bosdriesz. European Journal of Public Health, 24(suppl_2), cku164-043.
o Dai, H., & Hao, J. (2022). Online popularity of JUUL and Puff Bars in the USA: 2019–2020. Tobacco control, 31(1), 7-10.
· It’s not clear if the current paper uses Google trends data or a search of Google News. If the later, why not use Tobacco Watcher.
o Tobacco Watcher. Real time artificial intelligence for tobacco control. Available: https:// tobaccowatcher.globaltobaccocontrol.org/ [Accessed 12 Jun 2020].
· Lastly, the introduction lacks a clear purpose statement. What was the research aim or question?
Data and methods
· The information about Hilton et al.’s article in the first paragraph seems as though it might fit better in the introduction to help justify the use of online arguments about policy-related topic.
· The methods section current lacks important context describing the data collection process and procedures. It mentions that data were scraped. With what program did the authors scrap data, and did these procedures adhere to Google’s data sharing policy? Why did the authors not use google trends or google health trends API, or did they? Also, South Africa is not mentioned as a key term, how were searches restricted to onlin South African news?
· Figure 2 indicates that data were time-stamped, how were data reported – daily, weekly?
· Similarly, the qualitative coding procedures are minimally described. Explain the process that the authors used to develop a codebook. How many times did the authors reach the text before initial codes were identified? Did the authors use an inductive or deductive approach to developing codes?
· After all data were coded, it says they were randomly divided into three groups. Were data not double coded? If so, how was bias handled. If the authors did not originally double code data, I highly recommend they do this to ensure all data were coded correctly – reporting inter-rater reliability or agreement scores provided by NVivo? Also describe how discrepancies were handled.
· In the results section, there seems to be many other categories that were coded other than just these three themes. For example, figure 1 explains how articles were distributed, and table 1 reports stakeholders. Include text and perhaps a codebook to explain how codes were operationally defined and samples of text for each code. For example, what is eye witness news? Are there differences in reach between these sources?
· The results report context about the themes. Was a thematic analysis conducted after initial codes were developed? If so, explain methods used for analysis.
· Provide more explanation for how primary and secondary themes were determined. How was ‘most important’ determined – percentage of text coded in Nvivo? Also, move this explanation that is currently in the results to the methods.
· What program or software was used to calculate frequencies of reported codes.
Results
· Later in the results, it mentions that top 10 publishing houses. Are these reported in Figure 1? If so, state this here. Also, how are the authors determining the top 10 publishing houses? If these are determined via Google, perhaps that belongs in the methods.
· As mentioned above, the description of theme importance should be described in the methods section.
· It appears that there are sub-themes reported within each theme, similar to the sub-themes included for the economic/financial theme. What were some of the sub-themes for legal and medical? It would be helpful to include a table with the sub-themes along with the operational definition and sample text. Right now, it reads similar to a timeline; however, it is unclear how articles were selected to be reported per theme. Were there ‘peaks’ in discussions per theme that the authors are reporting in a particular timeline? Please provide more explanation in the methods and/or organize the results to show this.
· When it says, “the most common argument” on line 126, how often was this argument used? In what percentage of the articles was this mentioned?
· Some of the results read more like a discussion section. For example, Line 230-232, “However, at this point the judgement was moot as the tobacco sales ban had already been lifted on 18 August 2020.” And Line 262, “These cigarettes 262 were illegally sold in South Africa.” Provide a clear statement of ‘news articles describe’ or ‘articles depicted’ to demonstrate that the reported findings are from news articles and not the authors’ opinions.
· Line 183, what is well-publicized? How many articles discussed these legal challenges? This context would help explain the importance of these particular events.
Discussion
· In the first paragraph (line 304) it says that “the media generally reported negatively on the sales ban.” What percentage was this, and how was sentiment coded?
· Be careful with causal claims “Support for the sales ban decreased because:” This study appears to report media coverage about the topic, not public sentiment about the topic. Instead reframe that media portrayed sentiment to….
· The discussion reads like an extension of the results. What are the big take-aways from the findings? How can understanding media depictions about the sales ban help to inform future policy approaches?
· It is inappropriate to cite a paper that is under review. Remove self-citations that are under review.
· The paragraph about the illicit market is great.
· The paragraph from lines 334-338. Did this study measure lost revenue? If not, report that the news reported this – not your findings. Also, why is this topic relevant for tobacco control? Will all profits be lost, or will consumers spend in other areas?
· Other limitations to consider: were the search terms extensive? What bias were introduced due to the methodological approaches, and what measures did the authors use to control for this bias?
· The conclusion (lines 354-355) state, “While health advocates in South Africa have differing opinions around whether or 354 not the sales ban was a good idea…” Where was this measured or mentioned? Cessation was not mentioned in the paper, and data from health advocates does not appear to be collected. How did the findings support this claim? The medical theme section mentions the government ignoring nicotine withdrawal. The authors need to either make this connection (highlighting these findings in the discussion and discussing how prior research has addressed this) or remove this part.
· The final sentence of the discussion includes the same vague statement from the abstract. Please address using earlier comment from the abstract.
Minor comments
Abstract
· Line 20, “Because” should be capitalized. Also, did findings from the study suggest that the medical rationale was not well communicated?
· What do the authors mean by, “Any future tobacco control policy intervention should be well-considered”? Perhaps this was a government response to an issue that influenced tobacco control. If so, then the government could consider comprehensive approaches to tobacco control that restrict access to vulnerable groups while also providing cessation support to dependent users. Is this what the authors are trying to say?
Introduction
· Like 42, should this be “smoking lobbyist groups…”
· The authors state “Ebge, Ngobese, Barca, Crosbie” three times in three back-to-back sentences. Once is enough.
Data and methods
· Similar to the earlier comment, do not repeat the authors names over-and-over.
Results
· It may be a preference, but do not start sentences with a numerical value – line 107. If staring with a number, spell it out “i.e., Eight percent.”
· Line 119, remove “In fact.”
· Line 204, please rephrase to ‘multinational companies.’
· Medical and legal themes include the word theme in the title; however, economic/financial does not. Please stay consistent.
Discussion
· Line 342 is missing a period.
English is fine. There are a few typos, but they do not appear to be due to the quality of the English language.
Reviewer 2 Report
General
Authors explore in their manuscript ‘A media analysis of the Covid-19 tobacco sales ban in South Africa’
whether the prevalence and severity of depression and pain interference in married PD patients would differ from single PD patients as well as in persons free of PD. I like this topic as I think the area of caregivers is understudied, however some work should be done before this manuscript can be published.
Methods
please change <2020to> into ‘2020 to’
Introduction
the Introduction stops with the aim (This study places greater an emphasis on the economic aspects of the
sales ban, and covers a somewhat longer period from # to #.); please dont mention all kinds of methodological remarks in the Intro.
(so please remove the following:
We analyse the main arguments made by proponents and opponents of the tobacco sales ban. We focus on media articles published in English by South African media houses. Egbe, Ngobese, Barca, Crosbie 9 were the first to conduct a comprehensive content analysis of media coverage of the tobacco sales ban. Egbe, Ngobese, Barca, Crosbie 9 focussed on the process of the sales ban, and the role of the tobacco industry in undermining the government’s aims. Egbe, Ngobese, Barca, Crosbie 9 did not focus much on the economic aspects of the sales ban.)
bring these sentences to some other place
Methods
Sample
Please change:
<Focusing on South African media houses, we performed a Google search using the keywords smok*, puff*, lockdown, tobacco, and cigarette*, for articles published from 23 March 2020 (when the sales ban was announced) to 18 December 2020 (a week after the High Court ruled that the ban was unnecessary and unconstitutional). We started by scraping media reports from Google news, then scraped the largest news websites and finally smaller news outlets. The search yielded 1 512 potentially useful articles. We excluded duplicate articles (n=263), video articles (n=386), and articles from non-national media outlets (n=422). The final sample consisted of 441 articles (see Appendix).>
then go on with:
<We follow the approach of Hilton, Wood, Patterson, Katikireddi 10 who analysed the arguments made by ‘key claim-makers’ regarding the implementation of a alcohol minimum unit price in Scotland, based on a media analysis of 262 articles from ten UK national newspapers. Hilton, Wood, Patterson, Katikireddi 10 examined advocates’ and critics’ constructions of the alcohol problem and whether a minimum unit price would address the issue. They identified categories for a coding frame around a priori research questions.>
Please add an answer to the question: Did an Ethics Committee approve the study?
Measures
Importance: how did you measure this?
How did you come to the three: medical, legal, and economic/financial (if this comes from Hilton et al. please put this here)
Statistical analyses / Reporting
I suggest you to rewrite this part: First, we … . Next, we … . Then, we … . Finally, we … . The readers will easier grab what you did and in which order the Results will be presented.
Results
Tell me how it is possible, that the President abolishes the ban in August 2020 and that the High Court comes to a ruling in December 2020. Is not the case (after August 2020) loosing its essence?
Give me some answer to the following question: Can someone go directly to the High Court? (Yes: In its court case against the government, FITA; No: This led BATSA to file a lawsuit against the government at the Western Cape High Court)
Has S Africa undersigned the Framework Convention for Tobacco Control?
Please change <One of South Africa’s top scientists (Professor Shabir Madhi) said that ‘the damage caused by smoking occurs over a long period, and a short period of not smoking cannot undo long-term damage already done’ (Article 253), making the ban a bad idea.> into ‘One of South Africa’s top scientists (Professor Shabir Madhi) said that ‘the damage caused by smoking occurs over a long period, and a short period of not smoking cannot undo long-term damage already done’ (Article 253), making the ban a bad idea.’
Discussion
Most of the reviewers expect a lay-out of the Discussion like this
Para1 repeat the Aim; answer it (you now have the answer); avoid interpreting it
Para2-# please start with your own finding (we found …). Mentioning this finding will at the same time restrict the paragraph; in the second sentence start with previous publications. In the last sentence of this paragraph give some reason why the reader had to read all this.
Strengths and Limitations
Does this study have strengths?? Please start with this, and change also the heading. Limitations are sources of bias (selection bias, information bias) and how did you handle your confounders. What did these limitations mean for the findings?
Implications
Please add a paragraph on what your findings mean for practice, and what they mean for future research. Also add a heading.
Please rewrite parts of the discussion
It would be good to hear from you some supposed causal pathway.
Tables, Figures
Table 2: could not you make it in such a way, that the column Percentage Primary theme becomes part of the column Primary theme with between paracenteses (…) the numbers you calculated for Percentage Primary theme (without indicating every time that it is %)
References
2. Van Walbeek C, Filby S, van der Zee K. Lighting up the illicit market: Smoker’s responses to the cigarette sales ban in South Africa. 2020. [place??]
3. Van Walbeek C, Filby S, van der Zee K. Smoking and quitting behaviour in lockdown South Africa: Results from a second survey. 2020. [place??]
Round 2
Reviewer 1 Report
The authors have effectively addressed all of my comments.